# Improving Interpretability of Radiology Report-based Pediatric Brain Tumor Pathology Classification and Key-phrases Extraction Using Large Language Models

Chen Zhao* [†], Kareem Kudus* [‡ §], Sara Ketabi[† ‡], Khashayar Namdar[‡ §], Matthias W. Wagner [¶ ††], Birgit B. Ertl-Wagner[‡ § ¶ ‖], Farzad Khalvati[† ‡ § ¶ ‖ ** ‡‡]

*Equal Contribution
[†]Department of Mechanical and Industrial Engineering, University of Toronto, Toronto, ON, Canada
[‡]Neurosciences & Mental Health Research Program, Research Institute, The Hospital for Sick Children, Toronto, ON, Canada
[§]Institute of Medical Science, University of Toronto, Toronto, ON, Canada
[¶]Department of Diagnostic & Interventional Radiology, The Hospital for Sick Children, Toronto, ON, Canada
[‖]Department of Medical Imaging, University of Toronto, Toronto, ON, Canada
[**]Department of Computer Science, University of Toronto, Toronto, ON, Canada
[††]Institute of Diagnostic and Interventional Neuroradiology, University Hospital Augsburg, Germany
[‡‡]Correspondence: farzad.khalvati@utoronto.ca

*Abstract*—Radiology reports are crucial for bridging the expertise of radiologists and other clinicians. Machine Learning models trained on these reports have shown promising performance in various downstream clinical tasks, such as predicting the necessity of future follow-up procedures, based on past radiology reports. However, for clinicians to adopt these models and for radiologists to validate the results, interpretability of the model is essential. In this study, we train BERT models on radiology reports to classify pediatric brain tumor pathologies. These large language models enable accurate report-level classification, without the need for costly word-level annotations. To identify and extract keywords and key-phrases related to distinct pathologies from radiology reports, we used a modified Term Frequency-Inverse Document Frequency to determine phrase importance based on prevalence and attributions scores. We achieved an overall multiclass Area Under Receiver Operating Characteristic Curve (AUROC) of 79.57% using ClinicalBERT. Moreover, the per-class AUROC values were 86%, 71.2%, and 81.5%, for 'Pilocytic Astrocytoma', 'Low-Grade Astrocytoma', and 'Other' pathologies, respectively. Our explainability analysis identified hypotonia and mesencephalon as the most important terms for 'Pilocytic Astrocytoma' and 'Low-Grade Astrocytoma', respectively.

*Index Terms*—BERT, deep learning, keyword extraction, interpretability, key-phrase extraction, MRI, NLP, radiology reports

## I. INTRODUCTION

Radiology reports are considered as a primary form of communication between radiologists and their colleagues. These reports provide critical diagnostic information needed for informed decisions and thus, play a pivotal role in medical care [1]. However, due to the large amount of information clinicians must process, critical information is sometimes missed, leading to medical errors [2]. Healthcare outcomes could be improved by using Artificial Intelligence algorithms to extract important information from radiological reports more efficiently [3].

Natural Language Processing (NLP), is a research field that aims to have computers understand language. Traditional NLP approaches to key information extraction, such as part-of-speech (POS) tagging, rely on grammatical categories (noun, verb, etc.), assigned to each word in a given text. Performance of these traditional approaches is limited by a high false-positive rate, a result of their difficulty handling unknown words and different contextual meanings [4].

Large language models (LLMs) address the limitations of traditional NLP approaches; they are trained on massive text datasets and use billions of parameters to understand not only words but their context as well. LLMs, such as Bidirectional Encoder Representations from Transformers (BERT) [5] and Generative Pretrained Transformer (GPT) [6], have achieved remarkable success across various NLP tasks [7]. However, the adoption of LLMs in the medical domain has been limited due to their lack of interpretability, which has inspired many works aiming to enhance the transparency of LLMs. For example, Thomas et al. [8] showed that GPT-4 can be instructed to replicate the clinical reasoning patterns used by doctors when making a diagnosis. Healthcare professionals have more confidence in interpretable models in which they can identify potential biases and understand the factors driving predictions.

Tumor pathology impressions are oftentimes made by radiologists based on magnetic resonance imaging (MRI) examinations, with reasoning for the pathological prediction detailed in the radiology report. The gold standard for tumor pathology diagnosis assessment is the analysis of a tissue

sample. In this study, the objective was to predict tumor pathologies from radiology reports using LLMs, while also highlight key information in the report contributing to the prediction. Successful identification of tumor pathologies from radiology reports using LLMs would show these models have deep understandings of expert descriptions of radiological images. Highlighting key information would help readers of radiology reports more efficiently identify information relevant to the underlying pathology. Our interpretable model with a strong understanding of radiological reports could be used in the future for educational purposes, for example, to help new radiologists make sense of reports, or as a low-quality report detection tool, to identify cases where there is a mismatch between the radiologist's description of the image and the actual pathology.

## II. RELATED WORKS AND LITERATURE REVIEW

NLP aims to give computers the ability to understand text and spoken words. In radiology, NLP tools have been used for various tasks including information retrieval and text classification [9]. Traditionally, heuristic approaches such as dictionary and rule-based methods were used for NLP tasks in radiology. MedLEE, a natural language text extraction system, converts radiology reports into a structured format using a predefined dictionary [10]. Ontology-based clinical information extraction system [11] employs a rule-based approach to extract structured information from clinical notes.

More recently, machine learning (ML) techniques have been used for medical keyword extraction. The Mayo Clinical Text Analysis and Knowledge Extraction System (cTAKES) [12] combines dictionary and ML methods with the Unified Medical Language System (UMLS) [13] for dictionary inquiries, which involves mapping terms in clinical texts to standardized terms from a medical terminology database. ML methods such as POS tagging have been used to identify and classify medical terms. Wu et al. proposed a POS-based method to convert unstructured text data into structured reports that allow patient examination results to be more easily understood [14]. A Linear-chain Conditional Random Field (CRF) [15] is a type of statistical model that is used for sequence classification tasks such as POS tagging or named-entity recognition. Unlike models that classify each item in a sequence independently, Linear-chain CRFs take into account the entire sequence, helping them to excel in tasks where the context and order of items matter. Andrean et. al [16] employed a Linear-chain CRF for POS tagging and then constructed an information extraction system to retrieve and categorize clinical terms into nine groups from radiology reports.

In comparison to ML techniques, deep learning (DL) methods rely on neural networks that benefit from a larger number of parameters, allowing them to model more complex patterns in the data. DL models have been used for medical classification and keyword extraction. Charlene et al. [17] evaluated the effectiveness of various ML and DL techniques for the identification and assessment of ischemic stroke through MRI radiology reports. They found that DL methods, particularly those incorporating GloVe word embeddings [18] and Recurrent Neural Networks [19] demonstrated high accuracy.

Improving the interpretability of DL models in radiology is crucial for enhancing clinical trust and decision-making accuracy. William et al. [20] utilized a recurrent neural network (RNN) to generate descriptive text to explain model decision making process in human language, providing explanations that are easily understood by medical professionals.

## III. METHODOLOGY

All methods of this study were performed in accordance with the guidelines and regulations of the research ethics board of The Hospital for Sick Children (Toronto, Canada), which approved the study and waived informed consent. In this section, we describe our approach for classifying pediatric brain tumor pathologies from radiology reports and extracting key information. The workflow is illustrated in Fig. 1.

### A. Dataset and Pre-processing

We used radiology reports associated with pediatric brain MRI from The Hospital for Sick Children, which consists of 26415 MRI reports without labels, and 336 reports with a total of 15 different pathology labels acquired through histopathological analysis of tumor tissue collected during surgery. All reports are consistently structured, beginning with a summary of the patient's clinical history, followed by detailed observations from the MRI scans, and concluding with a diagnosis result from radiologists. All patients' personal information was removed from the reports to ensure data privacy. The labeled data was split into three categories: 144 reports classified under 'Pilocytic Astrocytoma', 67 under 'Low-Grade Astrocytoma', and 125 labeled as 'others'. Given the limited size of our labeled dataset, consisting of only 336 patients, and the uneven distribution of cases across 15 pathologies, we categorized the dataset into three classes to address potential class imbalance. Specifically, the two pathologies with the highest number of cases were designated as separate classes, while the remaining 13 pathologies were combined into a third class labeled as 'other'. This strategy was employed to create a more balanced distribution of labeled data, ensuring the model would be trained on approximately equal amounts of data from each class. By mitigating the risk of class imbalance, we aimed to prevent the model from disproportionately weighting certain classes due to uneven training data distribution. This approach allows the model to learn effectively and avoid biases that could compromise classification performance. We used the Radiology Lexicon from BioPortal [21], which contains radiological terms and their synonyms, to augment our dataset using synonym replacement. Before training the LLM, we removed irrelevant text such as headers, footers, and transcribed dates.

### B. Brain Tumor Pathology Classification

Our approach involved first pretraining LLMs on a large dataset of unlabeled data to learn semantic representations of radiology reports and then fine-tuning the pretrained models on labeled reports to identify pathology types. To ensure

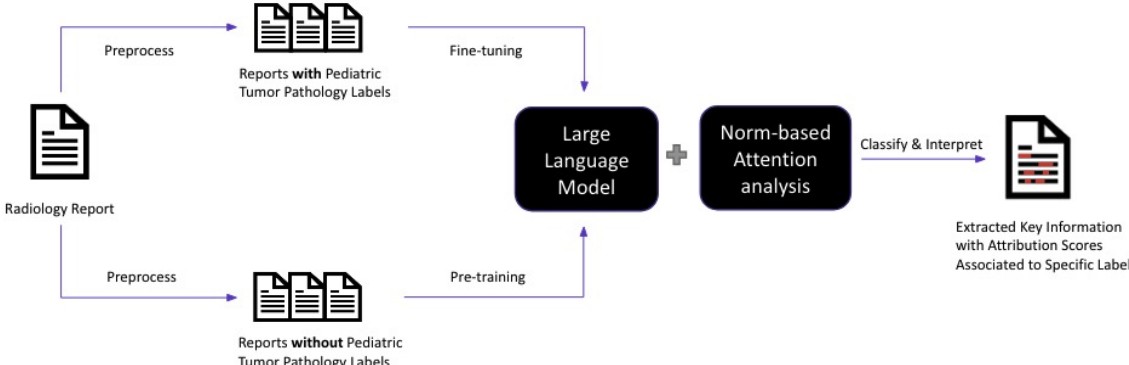

Fig. 1. Model Training & Interpretability Workflow: Two sets of Radiology reports are used to train the model: reports with and without pediatric tumor pathology labels. The preprocessed reports without labels are first used to pretrain the LLM; then the reports with labels are used for fine-tuning the model on the classification task. Norm-based attention analysis is applied to the fine-tuned model, resulting in attribution scores associated with classification labels for every word and phrase in each report.

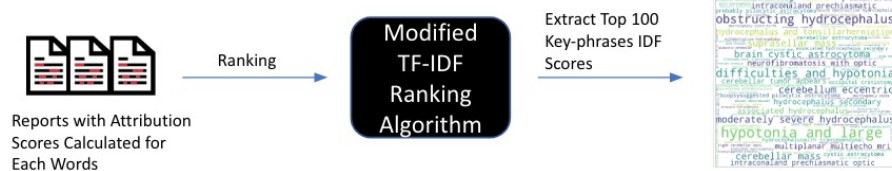

Fig. 2. Ranking Workflow: Illustration of the ranking workflow, focusing on how extracted key information is processed and ranked to identify the most relevant key-phrases among all reports. Reports with attribution scores calculated for each word and phrase, which are the output from the interpretability analysis, are fed into the modified TF-IDF ranking algorithm. This algorithm ranks the key-phrases, and the final output presents the top 100 key-phrases with their importance scores.

robustness of our results, we used 10-fold cross-validation. This is based on splitting the data into 10 subsets, training the model on 9 subsets and testing it on the remaining subset. This process was repeated 10 times, each time with a different subset as the test set. Performance metrics were averages across these 10 runs. As this study aims to extract informative keywords for each pathological class, the classification model serves a dual purpose. It not only categorizes reports into different pathological labels but also facilitates keyword extraction. The effectiveness of keyword extraction is directly linked to the accuracy of the classification model, where a high-accuracy model is more likely to be capable of understanding the text with more semantic meaning. Therefore, if an advanced classification model demonstrates superior performance, it would inherently improve our keyword extraction results by providing more accurate and relevant keywords. For this classification task, we relied on a set of state-of-the-art DL models, namely RadBERT [22], ClinicalBERT [23], and BERT-base-uncased [24].

1) **BERT-base-uncased:** BERT-base-uncased is a basic version of BERT, trained on a large corpus of text from Wikipedia and BooksCorpus. This extensive training allows BERT to capture a wide range of linguistic nuances and contextual information.

2) **RadBERT:** RadBERT is a variation of the BERT model that has been specifically fine-tuned on radiological data with initialization from BioBERT [25], a domain-specific language model pretrained on large-scale biomedical corpora.

3) **ClinicalBERT:** ClinicalBERT was trained on a large multicenter corpus of patients' records to fine-tune the base language model.

**Pretraining on Unlabeled Data**: The key method we used in the pretraining stage was masked language modeling [5], which is a fill-in-the-blank task, where a model uses the context words surrounding a mask token to predict what the masked word should be. We hypothesized this pretraining would help tailor the three BERT variants to data from our institution. The hyperparameters used for pretraining were: a learning rate of $1e-5$, batch size of 16, AdamW optimizer, and 10 epochs for training.

**Fine-tuning on Labeled Data**: After pretraining, we truncated the length of input text for each sample to 512 tokens, which is the maximum length accepted by BERT models and sufficient to retain the majority of symptom descriptions for most of the reports. We then fine-tuned the LLMs on the 336 labeled radiology reports for classification, splitting the reports into 302 training datapoints (90%) and 34 test datapoints (10%). During fine-tuning, the learning rate was set to $1e-5$, with a batch size of 8, using the AdamW optimizer and 20 epochs of training.

## C. Interpretability Techniques

DL models are often considered to be black boxes as their reasoning is hard to explain. Model interpretability is essential so that clinicians can trust and understand the predictions. Our research aims to improve LLM interpretability through key-phrase extraction from radiology reports. Specifically, we focus on a norm-based analysis of the attention mechanism [26] to identify key-phrases with high attributions to the classification process. Traditional interpretability techniques based on model weights analysis [27], [28] ignore input context. Norm-based attention analysis, in contrast, considers both model weights [29] and the input sequences.

The attention mechanism is a fundamental component of LLMs, enhancing their ability to understand and process complex input sequences [29]. At a high level, it helps the model focus on the most relevant parts of the input when making predictions. The attention mechanism relies on three components: query (the token of interest), key (all other tokens in the sequence), and value (the contextual information associated with each token). It computes weights that can be used as a measure of the relevance of each token in the input sequence to the target class. The process, referred to as self-attention, generates a vector $y_i$ for each input token $x_i$, which captures the token's contextual information within the input sequence. These embeddings are then aggregated and passed through subsequent layers, such as classification layers for the final prediction. For each input vector $x_i$, the attention weights $\alpha_{i,j}$, in Eq. 1, are calculated based on the inner product between the query $q(x_i)$ and the key $k(x_i)$ of each token, scaled by $\sqrt{d'}$ where $q(\cdot)$, $k(\cdot)$, and $v(\cdot)$ are the Query, Key, and Value vectors, respectively, and $\mathcal{X} = x_1, \ldots, x_n \subseteq \mathbb{R}^d$ represents the sequence of input vectors. $d$ and $d'$ represent the dimensions of the inputs and attention space, respectively, while $n$ is the number of input tokens.

$$\alpha_{i,j} := \text{softmax}_{x_i, x_j \in \mathcal{X}} \left( \frac{q(x_i) k(x_j)^\top}{\sqrt{d'}} \right) \in \mathbb{R}^{n \times n} \quad (1)$$

Here, $\mathbf{x}_i$ and $\mathbf{x}_j$ correspond to the i-th and j-th token representations within the encoder stack. The vector $\mathbf{y}_i$ is updated by interactions with all $\mathbf{x}_j$ vectors, where every vector is a representation of a specific token in the radiology reports. This self-attention mechanism allows each token to dynamically influence the contextual embedding of every other token.

The value vectors represent the actual information from the input sequence that will be passed on to following model processes after being scaled by attention scores. Similar to how query and key vectors are generated, the value vectors $v(x_j)$, in Eq. 2, are obtained by transforming the input vectors using the value transformation where all $W^Q$, $W^K$ and $W^V$ $\in \mathbb{R}^{d \times d'}$, all $b^Q$, $b^K$ and $b^V \in \mathbb{R}^{d'}$, and $d'$ equals to $d$

$$\begin{aligned} q(x_i) &:= x_i W^Q + b^Q \\ k(x_j) &:= x_j W^K + b^K \\ v(x_j) &:= x_j W^V + b^V \end{aligned} \quad (2)$$

With the attention weights derived, the output vector $y_i$ in Equation 3 is calculated as a weighted sum of the value vectors $v(x_j)$. To ensure that the output $y_i$ has the same dimension as the input $x_i$, we apply a linear transformation using $W^O \in \mathbb{R}^{d' \times d}$. Since we're setting $d' = d$ (i.e., the attention dimension equals the input dimension), $W^O \in \mathbb{R}^{d \times d}$.

$$y_i = \left( \sum_{j=1}^{n} \alpha_{i,j} v(x_j) \right) W^O. \quad (3)$$

To interpret the LLM's decision-making process, we leverage Equation 3, which shows that the attention mechanism effectively computes a weighted sum of the transformed input vectors, denoted as $f(x)$, where:

$$f(x) = \left( x W^V + b^V \right) W^O. \quad (4)$$

Therefore, the output vector $y_i$ can be expressed as:

$$y_i = \sum_{j=1}^{n} \alpha_{i,j} f(x_j). \quad (5)$$

We can then compute the Euclidean norm of each weighted, transformed vector $\alpha_{i,j} f(x_j)$:

$$A_{i,j} = ||\alpha_{i,j} f(x_j)||, \quad (6)$$

where $||.||$ denotes the Euclidean norm in $\mathbb{R}^d$. By analyzing the values of $A_{i,j}$, we gain valuable insights into the LLM's attention during pathology classification. This interpretability technique helps us understand how the model processes and weighs different words, ultimately quantifying the keyword attributions and facilitating further extraction processes from radiology reports.

## D. Keyword Extraction

A popular method for key-phrase ranking is the Term Frequency-Inverse Document Frequency (TF-IDF) algorithm which is a numerical statistical measurement used in information retrieval and text mining to evaluate the importance of a word within a collection of documents. Traditionally, TF-IDF uses the frequency of words to determine how relevant those words are to a given document. The importance of a term is determined by TF and IDF. TF measures the frequency of a term in a report, while IDF assesses the rarity of the term across the entire collection of documents. The product of TF and IDF evaluates the significance of each term.

Our method extends the traditional TF-IDF approach by replacing the TF scores with the attribution scores obtained from the norm-based attention analysis method (Fig. 2). These scores are derived from the model's attention mechanism, which highlights how much each part of the input data influences the model's predictions. In our case, attribution scores are numerical values that quantify the contribution of each word or phrase in a radiology report to the model's decision. We used the same equation as TF-IDF but replaced TF with attribution scores.

Each individual token in the radiology reports has corresponding attribution scores. We calculated a modified TF-IDF weight for each term, which takes into account these attribution scores. By doing so, single keyword or phrase attribution can be calculated across all reports. This can further rank the words or phrases among all reports based on model prediction rather than merely word occurrence. The modified TF-IDF weight for a specific term $i$ that may occur multiple times in a report is given by Equation 7.

$$\text{Modified TF-IDF}_i = \sum_{j=1}^{n_i} \left( \text{Attribution Score}_{ij} \times \text{IDF}_i \right) \quad (7)$$

In this equation, $n_i$ represents the total number of occurrences of term $i$ in the radiology report, Attribution Score$_{ij}$ denotes the attribution score for the $j$-th occurrence of term $i$, obtained from the norm-based attention analysis, and IDF$_i$ is the inverse document frequency for term $i$ calculated based on all documents. Our IDF scores are calculated using all radiology reports. When a word or phrase appears in many reports, its IDF score decreases, as it is considered common and less informative. Conversely, a term that appears in fewer reports receives a higher IDF score, indicating it provides more unique information about the reports it appears in.

By using attribution scores instead of the raw term frequencies, we aim to enhance the keyword extraction process with more contextually relevant keywords.

The modified TF-IDF algorithm includes six modules.

1) Calculating the Attribution Scores: We compute the attribution scores for each token in the radiology reports according to their contributions to model classifications.
2) Handling sub-tokens: Due to the tokenization process, words can be split into sub-tokens to handle tokens unseen in our tokenizer. For example, 'sinuses', which is not in pre-defined tokenizers, is separated into 'sin' and '-uses'. Thus, a leading hyphen in a token indicates an original token was split into two sub-tokens, ensuring the procedure is reversible. For each word split into sub-tokens, we aggregate their attribution scores by averaging the scores associated with each sub-token for that word. This ensures attribution scores correspond to words instead of sub-tokens.
3) Computing the modified TF for term $i$: We replace raw term frequency of term $i$ with word-level attribution scores calculated from step 1.
4) Calculating the IDF for term $i$: The IDF remains unchanged and is computed based on the rarity of of term $i$ across the entire collection of radiology reports.
5) Computing the Modified TF-IDF Weight for term $i$: The modified TF-IDF weight for term $i$ is obtained by multiplying its attribution score with the IDF value.
6) Keyword Extraction: We rank all terms based on their modified TF-IDF weights and extract the top keywords as the most relevant terms among all reports for each specific pathology label.

*E. Phrase Extraction*

We further explore phrase extraction to identify meaningful multi-word phrases (mainly bi-grams and tri-grams) from radiology reports. Phrases contain more contextual information compared to individual words, which is particularly important in the medical field where complex phrases are used to convey details about medical conditions, procedures, and findings.

Our approach to phrase extraction includes modules similar to keywords extraction.

1) Calculating the Attribution Scores: Attribution scores for each token are computed in the same way as in keyword extraction
2) Handling sub-tokens: The phrase extraction process involves analyzing groups of words rather than treating individual words in isolation. Words in a phrase may be decomposed into subtokens, depending on whether they exist in the model's token set. To determine the attribution of an entire phrase, the attribution scores of all subtokens corresponding to the words within the phrase are averaged. For instance, if a phrase contains two adjacent words, and each word is split into two subtokens, the attribution for the phrase would be calculated by averaging the attribution scores of the four subtokens. This method, combined with a comprehensive token set for the model, ensures that the original medical terms are preserved by attributing importance to the entire phrase as a unit, rather than focusing solely on individual subtokens. By averaging the attribution scores across all subtokens, the integrity and meaning of complex medical terms are maintained, allowing for a more accurate and contextually relevant interpretation of medical terminology.
3) Filtering of Stopword Phrases: To enhance the relevance of the extracted phrases, we remove multi-word phrases that start or end with stopwords which is a collection of commonly used words.
4) Ranking and Selection: We rank the phrases based on their weight obtained by applying our proposed modified TF-IDF Weight. The top-ranked phrases are selected as contextually relevant and meaningful multi-word expressions in the radiology reports.

## IV. Experiments and Results

**Area Under Receiver Operating Characteristic Curve (AUROC) Scores**: Table I shows the averaged AUROC scores for different BERT models. The ClinicalBERT model performed the best overall with pretraining, resulting in an AUROC of 0.784. This model achieved an AUROC of 0.86 for 'Pilocytic Astrocytoma', 0.712 for 'Low-Grade Astrocytoma', and 0.815 for 'others'. Note that even though ClinicalBERT consistently outperformed the other BERT variants and pretraining seemed to help, the difference between the performance of the various models and the effects of pretraining seemed to be minimal, given the high standard deviations.

A detailed examination of the AUROC scores reveals that without pretraining, ClinicalBERT achieved the highest per-

formance with AUROC scores of 0.872 for 'Pilocytic Astrocytoma', 0.664 for 'Low-Grade Astrocytoma', and 0.809 for other pathologies. RadBERT and BERT-base-uncased models performed slightly lower in comparison. With pretraining, ClinicalBERT continued to outperform the other models, achieving AUROC scores of 0.860 for 'Pilocytic Astrocytoma', 0.712 for 'Low-Grade Astrocytoma', and 0.815 for other pathologies. The RadBERT and BERT-base-uncased models also showed improvements with pretraining, though not as significantly as ClinicalBERT. This suggests that while domain-specific pretraining is beneficial, the overall impact may not be as substantial as initially expected.

**Word Clouds**: Fig. 3 displays two word clouds representing the top 100 extracted keywords for the two most frequent tumor pathologies in our datasets (left for 'Pilocytic Astrocytoma', right for 'Low-Grade Astrocytoma'). The size of each word corresponds to its modified TF-IDF score, aiding in the identification of key terms associated with each pathology.

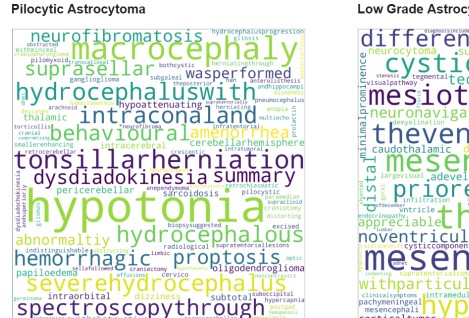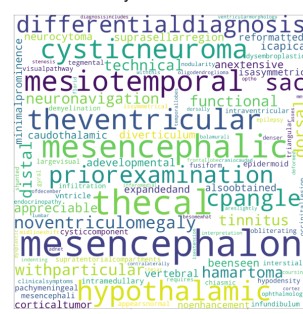

Fig. 3. Top 100 Keywords for 'Pilocytic Astrocytoma' (left) and 'Low-Grade Astrocytoma' (right). The word sizes reflect modified TF-IDF scores.

Fig. 4 presents two word clouds representing the top 100 extracted phrases for specific tumor pathologies (left for 'Pilocytic Astrocytoma', right for 'Low-Grade Astrocytoma'). The size of each phrase indicates its significance across all reports, providing a clear representation of key-phrases associated with these tumor types.

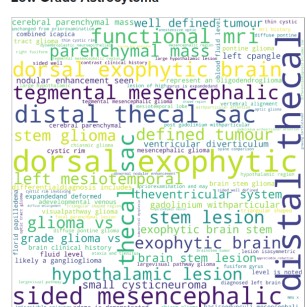

Fig. 4. Top 100 Key-phrases for 'Pilocytic Astrocytoma' (left) and 'Low-Grade Astrocytoma' (right). Phrase sizes reflect significance across the reports.

We analyzed the top 30 words and phrases for 'Low-Grade Astrocytoma' and 'Pilocytic Astrocytoma'. It was found that extracted phrases generally provided more insights into

specific pathology-relevant information compared to individual words. For 'Low-Grade Astrocytoma,' only one phrase was irrelevant, whereas three words were irrelevant for this class. In the case of 'Pilocytic Astrocytoma', five words were found to be irrelevant. One primary ongoing challenge is the typographical errors and misspelling, specifically the improper handling of spaces between words. For example, 'Theventricular' is supposed to be two words of 'The' and 'ventricular'. Efforts to resolve this issue could be one of the future works to enhance the accuracy of our text extraction processes.

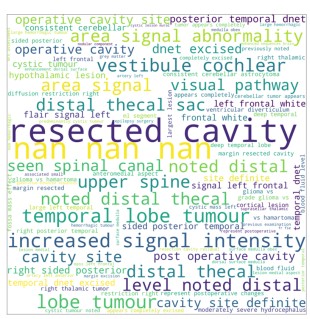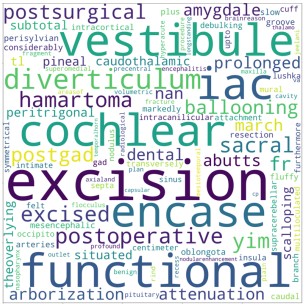

Fig. 5. Top 100 Key-phrases (Left) and Keywords (Right) Extracted by TF-IDF

In Fig. 5, we compared our proposed pipeline with conventional TF-IDF. The results showed that TF-IDF extracts words and phrases without accounting for pathology types. Thus, our method, tailored to specific pathologies, offers more detailed insights that enhance decision-making in clinical settings.

**Top 10 Extracted Key-phrases**: Table II shows the most frequent phrases and terms corresponding to the 'Low-Grade Astrocytoma' and 'Pilocytic Astrocytoma' classes.

**Example of keyword importance**: An example of keyword importance is visualized in Fig. 6. The darker color implies higher importance for pathology classification and vice versa.

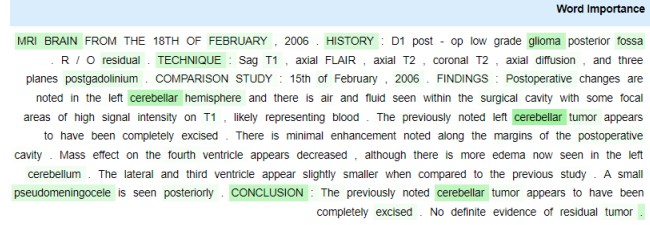

Fig. 6. Keywords Visualization for An Example of 'Pilocytic Astrocytoma'

## V. DISCUSSION

Rule-based and traditional keyword extraction methods can be effective for general NLP but may lack the necessary level of adaptability in specialized fields such as medicine. Most ML and DL keyword extraction methods require word-level annotations, which come with a high labeling cost. Additionally, both traditional and ML methods primarily focus on extracting nouns [12], [30], potentially omitting other crucial information, especially when dealing with complex medical narratives or ambiguous terminology.

TABLE I
AVERAGED AUROC SCORES AND STANDARD DEVIATION FOR DIFFERENT BERT MODELS

| Model | Pilocytic Astrocytoma | Low-Grade Astrocytoma | Others | Overall |
|---|---|---|---|---|
| *Without pretraining* | | | | |
| RadBERT | $0.829 \pm 0.028$ | $0.621 \pm 0.102$ | $0.750 \pm 0.008$ | $0.733 \pm 0.046$ |
| ClinicalBERT | $\mathbf{0.872 \pm 0.041}$ | $\mathbf{0.664 \pm 0.076}$ | $\mathbf{0.809 \pm 0.019}$ | $\mathbf{0.782 \pm 0.045}$ |
| BERT-base-uncased | $0.855 \pm 0.025$ | $0.613 \pm 0.089$ | $0.792 \pm 0.022$ | $0.753 \pm 0.045$ |
| *With pretraining* | | | | |
| RadBERT | $0.856 \pm 0.041$ | $0.678 \pm 0.059$ | $0.803 \pm 0.028$ | $0.779 \pm 0.042$ |
| ClinicalBERT | $\mathbf{0.860 \pm 0.033}$ | $\mathbf{0.712 \pm 0.114}$ | $\mathbf{0.815 \pm 0.026}$ | $\mathbf{0.784 \pm 0.057}$ |
| BERT-base-uncased | $0.853 \pm 0.038$ | $0.635 \pm 0.096$ | $0.782 \pm 0.050$ | $0.757 \pm 0.061$ |

TABLE II
TOP 10 FREQUENT TERMS RELATED TO 'LOW-GRADE ASTROCYTOMA' AND 'PILOCYTIC ASTROCYTOMA' CLASSES

| Low-Grade Astrocytoma Class | | Pilocytic Astrocytoma Class | |
|---|---|---|---|
| **Term** | **Score** | **Term** | **Score** |
| mesencephalon | 0.0126 | hypotonia | 0.0250 |
| distal thecal | 0.0119 | severe hydrocephalus | 0.0114 |
| thecal | 0.0109 | hypotonia and large | 0.0107 |
| dorsal exophytic | 0.0105 | macrocephaly | 0.0103 |
| sided mesencephalic | 0.0101 | tonsillar herniation | 0.0099 |
| distal thecal sac | 0.0101 | obstructing hydrocephalus | 0.0099 |
| mesencephalic | 0.0095 | difficulties and hypotonia | 0.0092 |
| thecal sac | 0.0087 | hydrocephalus with | 0.0084 |
| mesiotemporal | 0.0074 | moderately severe hydrocephalus | 0.0082 |
| theventricular | 0.0074 | cerebellum eccentric | 0.0081 |

LLMs, a subset of DL, have the potential to remedy the limitations of traditional and ML-based NLP methods for medical applications. We trained LLMs on radiology reports to identify brain tumor pathologies. Our adaptation of the traditional TF-IDF framework integrates attribution scores derived from norm-based attention analysis. This method quantifies the influence of specific text elements on model decisions. In contrast, earlier approaches focus on generating descriptive text using recurrent neural networks (RNNs) as an interpretability technique. Such approaches provide explanations that are intuitive and easy to understand for clinicians but may introduce inaccuracies or misleading information since language models are prone to hallucination. Our proposed method minimizes the risks of hallucination by focusing on the direct contributions of text elements contained in the report.

With an overall AUROC score of nearly 80% for our best model, there is strong evidence that our pipeline can accurately identify pathology from radiological reports. Furthermore, the results (Fig. 3, Fig. 4, Fig. 6) illustrate that our key-phrase extraction approach was able to highlight medically relevant terms in radiological reports. The word clouds (Fig. 3, Fig. 4) offer a straightforward view of relevant keywords and phrases associated with different pathologies. These could be highlighted for healthcare professionals to facilitate more efficient report analysis. The focus on two specific pathology labels in this study is primarily due to the current scale of our dataset, as discussed in Section III-A. Our framework is capable of being expanded directly to other related problems, including other medical conditions and multi-class problems with sufficient data. Nothing about our approach is specific to the use case in this paper and thus can be applied widely for improved keyword identification in any radiology reports.

Despite the promising outcomes, our study has several limitations. These include the restricted number of labeled data points and errors in the data such as incorrect spelling. To address these limitations, future work will focus on several key areas. First, increasing the dataset size will help mitigate overfitting and enhance the model's generalizability and robustness. Second, employing additional data augmentation techniques will correct misspellings and improve data quality. Finally, experimenting with larger language models, such as GPT models, which are more capable of handling noisy data and extracting contextual information, may reduce the impact of typographical errors and improve overall model performance. In this study, we conducted a three-class classification, distinguishing between 'Pilocytic Astrocytoma', 'Low-Grade Astrocytoma', and a collective group of other pathologies. Future research should explore more granular classification using a greater number of pathology classes and larger sample sizes to improve model performance and clinical relevance.

## VI. CONCLUSION

Our study explored the potential for LLMs to identify pediatric brain tumor pathologies from radiology reports, while highlighting key-phrases for interpretability purposes. To that end, we integrated norm-based attention analysis to identify key-phrases contributing to the tumor pathology classifications. Our models, trained on report-level labels, achieved strong classification performance. Furthermore, they efficiently extracted relevant key-phrases without relying on costly and laborious word-level annotations used in traditional keyword extraction methods.

ACKNOWLEDGMENT

This work was supported by the Chair in Medical Imaging and Artificial Intelligence, a joint Hospital-University Chair between the University of Toronto, The Hospital for Sick Children, and the SickKids Foundation.

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
