# OpenReview forum: "Improving Interpretability of Radiology Report-based Pediatric Brain Tumor Pathology Classification and Key-phrases Extraction Using Large Language Models"
_IEEE.org/EMBS/BHI/2024/Conference — IEEE BHI'24_

### Official Review · Reviewer_mhDT · 2024-08-03
**Improving Interpretability of Radiology Report-based Pediatric Brain Tumor Pathology Classification and Key-phrases Extraction Using Large Language Models**

**Overall Rating:** 8
**Confidence:** 5

**Other Quality Metrics:**

(a) Clarity of writing: Good
(b) Clinical Significance: Great
(c) Methodological Novelty: Excellent
(d) Experiments and Results: Good

**Questions For The Authors:**

n/a

**Strengths:**

The work demonstrates several key strengths, particularly in terms of improving translation into clinical applications of BERT-based models. By incorporating criterion-based psychoanalysis to extract and establish key claims, the analysis increases model transparency, making AI predictions more meaningful and feasible highly reliable for healthcare professionals This is crucial for real-world adoption of AI in medical settings.

Another strength is a new approach to key phrase extraction using a modified TF-IDF algorithm that uses attribution scores. This approach goes beyond traditional methods and yields more contextual and meaningful sentences, which can greatly improve the efficiency of information retrieval and support clinical decision making.

The strong performance of the ClinicalBERT model, especially its high AUROC across classes, further underscores the power of domain-specific language models in clinical practice. This, together with the improved interpretive capabilities, makes it a promising tool for providing diagnostic radiology accuracy and efficiency. The generalizability of the developed methods, which can be adapted to other clinical settings, enhances the overall impact of the project, allowing it to be applied across clinical settings.

**Summary Of The Paper:**

The paper focuses on the use of BERT-based large languages ​​(LLMs) to classify pediatric brain tumor pathology from radiology reports, with greater emphasis on improving model interpretation for clinical validation Studies 26 , Using a database of 415 unlabeled and 336 labeled pediatric brain MRI reports, categorized into pilocytic astrocytoma, low-grade astrocytoma, and three other models—RadBERT, ClinicalBERT, and BERT-base-uncased— . used, and evaluated by ClinicalBERT as 79.57 The highest overall AUROC of % was obtained. Models are refined using 10-fold cross-validation to ensure robustness.

The authors enhance interpretation by applying focus analysis to LLM, which results in a modified TF-IDF algorithm to extract and rank key terms important for the classification process , including concept-based attribution scores to improve the significance of extracted keywords is Presented word clouds and disease-specific keywords, demonstrating the model’s ability to build related clinical emphasis.

This study makes a valuable contribution by showing that BERT-based models can correctly classify brain tumor pathology from radiology reports, and the paper also provides a new approach to improve interpretation if does not require an explanation that acknowledges labor It acknowledges limitations, such as a limited number of labeled data points and data quality issues, and suggests future work to expand the dataset, improve data quality, and explore language patterns sizes for increased efficiency.

**Weaknesses:**

A minor weakness of the study is the limited size of the data set, in particular the small number of labeled reports on air conditioning (336 in total) used to refine the models. This limitation may also affect the ability of the model to address the variability and complexity of medical language across institutions and patient populations. Expanding the dataset or adding additional data sources can increase the robustness and reliability of the model predictions in a real-world setting.

---

### Official Review · Reviewer_QPs9 · 2024-08-09
**This paper proposes an interesting concept, but should improve clarity of writing and clarify some of the methodological choices**

**Overall Rating:** 7
**Confidence:** 3

**Other Quality Metrics:**

(a) Clarity of writing: fair
(b) Clinical Significance: good
(c) Methodological Novelty: good
(d) Experiments and Results: Fair

**Questions For The Authors:**

* For the modified TF-IDF, is the Inverse Document Frequency calculated with all of the radiology reports, or did you exclude reports which had the same classification as the report which is currently being assessed?

* In section III.D, module (3) refers to a "modified TF". What is the modified TF? I was under the impression that the TF was being replaced entirely by the attribution scores, which in this list are calculated in module (1).

* Why is section "IV. Experiments" present and empty?

* This paper specifically delves into learning the relevant words for 2 conditions, how does this expand to a general framework for highlighting key words in any radiology report? I think that addressing this question could greatly improve the clinical significance portrayed in this work.

**Strengths:**

The particular application of NLP/LLMs in healthcare being addressed in this paper, improving communications between healthcare professions by reducing the amount of effort it takes to interpret each other's reports, has the potential to improve healthcare outcomes. The results of this paper demonstrate that the alteration to a clinician's workflow could be as simple as highlights of key information; this type of implementation has very low risk for causing medical errors, and appears like it could only be beneficial.
In method proposed in this paper, scaling of attribution scores through a modified TF-IDF approach, seems like an intuitive way to judge the relevance of a word/phrase for the final classification. It ensures that a word/phrase is relevant, without being common boilerplate language.

**Summary Of The Paper:**

In this paper, large language models were fine-tuned to classify radiology reports of pediatric MRI scan into one of three categories: pilocytic astrocytoma, low grade astrocytoma, or "other". The ground-truth labels were acquired from histopathological analyses. Additionally, this paper proposes a modified "term frequency-inverse document frequency" algorithm for determining the words/phrases that were most important for the classification.

**Weaknesses:**

* The scope and motivation of the paper can be confusing throughout the paper. While it is clear from the introduction that the goal is to "extract important information from radiological reports more efficiently", some of the wording in later sections switches the narrative unexpectantly. For example, in section III.B, "Our primary objective was... to classify tumor pathology into one of three categories", which can be confusing since it isn't apparent how this addresses the overall goal. In short, the ways in which each step are directly building towards the ultimate goal of the paper is hard to follow within the sections.
* I believe that section III.A is too short of a description of the dataset. It is not clear why the labeled data was split into the three specific categories that were chosen. It is also not clear why there are only 15 different pathology labels, or how they were reduced into the three categories. Were these conditions of particular interest for some reason? What is represented by "Others" and why is this category so large?
* The results of the modified TF-IDF were not compared to regular TF-IDF or other keyword extraction techniques. An additional experiment and at least a brief mention in the discussion would validate the use of LLMs, and show whether their results are actually better than the baseline algorithms.

---

### Official Review · Reviewer_sFUz · 2024-08-12
**Improving Interpretability of Radiology Report-based Pediatric Brain Tumor Pathology Classification and Key-phrases Extraction Using Large Language Models**

**Overall Rating:** 6
**Confidence:** 4

**Other Quality Metrics:**

(a) Clarity of writing: Good
(b) Clinical Significance: Fair
(c) Methodological Novelty: Good
(d) Experiments and Results: Good

**Questions For The Authors:**

Overall, the manuscript is well-written and presents significant findings. However, addressing the specific concerns outlined above in greater detail would enhance the clarity and robustness of the manuscript.

**Strengths:**

The manuscript contributes to enhancing the explainability of large language models (LLMs) in the classification of pediatric brain tumor pathologies. The paper is well-structured, with a clear and thorough articulation of the methods employed. The paper demonstrates a method to enhance the interpretability of BERT-based models by introducing a novel key phrase extraction method. The integration of modified TF-IDF with attribution scores provides a more nuanced understanding of how specific phrases contribute to classification, aligning the model's outputs with clinical reasoning and enhancing its potential for real-world application in medical settings.

**Summary Of The Paper:**

This study presents a novel approach to enhancing the interpretability of BERT-based models for classifying pediatric brain tumor pathologies in radiology reports. By integrating a modified Term Frequency-Inverse Document Frequency (TF-IDF) approach, which replaces traditional term frequency scores with attribution scores derived from norm-based attention analysis, the method identifies and ranks key phrases crucial to the classification process. The model achieved a multiclass AUROC of 79.57%, demonstrating its potential in clinical applications. The key phrase extraction approach effectively highlights medically relevant terms, as illustrated in the word clouds, offering healthcare professionals a clear and efficient way to analyze and interpret report findings.

**Weaknesses:**

The authors aim to classify tumor pathology into three categories: Pilocytic Astrocytoma, Low-Grade Astrocytoma, and Other. However, the rationale behind selecting these specific categories is unclear. What justifies the focus on these three classifications, and how does it impact the model's generalizability and clinical relevance?

The method describes aggregating sub-tokens to form original phrases, but the process of identifying these phrases is not thoroughly explained. How were the phrases identified, and what criteria were used to ensure that the aggregated phrases accurately represent the original medical terminology? These should be mentioned to improve reader understanding.

The model identifies certain keywords as more frequent in one category than in others. Is there a physiological explanation for this distribution, and does it align with clinical understanding? Providing a clinician-validated explanation would enhance the reliability and credibility of the model’s outputs.

Were there other studies that applied deep learning (DL) for this specific task of tumor pathology classification using radiology reports? How does the performance of this model compare to those in terms of accuracy, interpretability, and clinical applicability? Understanding its relative strengths and weaknesses would offer valuable context for its contribution to the field.

The authors state that their method extends the traditional TF-IDF approach by replacing TF scores with attribution scores derived from a norm-based attention analysis. They claim to be the first to adapt TF-IDF for interpretability using attribution scores. What are some other methods that have been explored for enhancing interpretability in similar contexts? What are the strengths and weaknesses of your method compared to those other approaches, and why might your method perform better in this particular context? Those should be mentioned to enhance the manuscript further.

---

### Decision · Program_Chairs · 2024-09-23

Accept